# A new system of phosphorus and calcium requirements for lactating dairy cows

**André Soares de Oliveira**[ID]*, **Suziane Rodrigues Soares**

Dairy Cattle Research Laboratory, Universidade Federal de Mato Grosso, Sinop, Mato Grosso, Brazil

* andre.oliveira@ufmt.br

## Abstract

Accurately predicting phosphorous (P) and calcium (Ca) dietary requirements is critical for optimizing dairy cattle performance, and minimizing mineral excretions and ecosystems eutrophication. This study provides a new factorial system to determine net and dietary P and Ca requirements for maintenance and lactation, derived from a meta-regression of mineral trials involving lactating dairy cows. A comprehensive global database was constructed from 57 peer-reviewed articles of mineral balance trials, with a wide range of dietary and animal performance data. We estimated the net requirements for maintenance from the intercept of a nonlinear equation between mineral intake and the sum of total fecal and urinary excretions, which is an estimate of endogenous mineral loss. Mineral secreted in milk was used to obtain net requirements for lactation. The mineral metabolizable coefficient was quantified through observed (treatment means) mineral intake and total fecal and urinary excretions, discounting the estimated endogenous excretions from our proposed models. The nonlinear models of total fecal and urinary mineral excretion were evaluated (observed versus predicted values) using a 5-fold cross validation approach. The models to estimate the sum of endogenous fecal and urinary excretions of P ($0.135_{\pm0.043}$ g P/kg BW$^{0.75}$) and Ca ($0.360_{\pm0.144}$ g Ca/kg BW$^{0.75}$) exhibited suitable precision and accuracy; r = 0.89 and 0.79, concordance correlation coefficient = 0.85 and 0.77, and root mean square prediction error = 24.1 and 20.5% observed means, respectively. Dietary variables (forage level, fiber, starch, crude protein, and ether extract) did not affect the metabolizable coefficient (MC) of P and Ca; therefore, an overall dietary MC of P ($0.69_{\pm0.01}$) and Ca ($0.65_{\pm0.02}$) were proposed. Our new system estimates lower net and dietary P requirements for lactating dairy cows compared to the NASEM-2021 and NRC-2001 models, but slightly higher Ca requirements than NASEM-2021. This proposed system holds potential to reduce the use of phosphorus in diets for dairy cows, and thus to enhance economic efficiency and environmental sustainability of the dairy industry.

## Introduction

Phosphorus (**P**) and calcium (**Ca**) are the most abundant minerals in the animal body [1], and two of the more abundant minerals in milk [2]. Inorganic sources of P and Ca represent the

**Data Availability Statement:** The data are held in a public repository. The complete dataset file is available from the Mendeley Data database (Soares, Suziane Rodrigues; Oliveira, André Soares de (2024) "Complete dataset of phosphorus and

calcium balance trials used to develop the mineral requirement submodel of The Nutrition System for Dairy Cattle. Dairy Cattle Research Lab, Universidade Federal de Mato Grosso, Campus Sinop, Brazil.", Mendeley Data, V2, doi: 10.17632/8t6f7229r4.2; https://data.mendeley.com/datasets/8t6f7229r4/2) All other relevants data and the inputs/codes/ioutputs of the statistical analysis are with the manuscript and its Supporting Information Files.

**Funding:** This study was funded by the Coordenação de Aperfeiçoamento de Pessoal de Nível Superior (CAPES; Brazil; scholarship of master's degree in Animal Science for Suziane Rodrigues Soares at the Universidade Federal de Mato Grosso – Campus Sinop; 2016-2018), Conselho Nacional de Desenvolvimento Científico e Tecnológico (CNPq, Brazil; Number: 309450/2019-5) and Ministério Público do Estado de Mato Grosso (Fundação Uniselva/UFMT. Brazil; Number SEI 23108.066569/2023-30). The funders had no role in study design, data collection and analysis, decision to publish, or preparation of the manuscript.

**Competing interests:** The authors have declared that no competing interests exist.

most expensive minerals supplemented in dairy cattle diets per animal. Moreover, excessive excretions from P overfeeding can contribute to soil and aquatic ecosystem eutrophication [3, 4]. Therefore, accurately predicting P and Ca requirements is critical for optimizing dairy cattle performance, economic efficiency, and environmental sustainability.

Mineral requirements are typically estimated through a factorial approach, and then evaluated or refined through response-dose feeding trials. Dietary requirements of minerals are computed by dividing the total net requirements by the mineral diet true absorption coefficient [5] or retention coefficient [6]. The concept of the true absorption coefficient (AC) is appropriate for minerals when only obtained from total and endogenous fecal excretions [5]. However, when the total and endogenous urinary excretions are also factored into the calculation, we proposed here the term metabolizable coefficient (MC) opposed to AC.

Lactation and maintenance are the major components of the net mineral requirement for lactating dairy cows. The net mineral requirement for lactation represents the amount of mineral secreted in milk, and it is relatively straightforward to obtain. The net mineral requirement for maintenance represents the sum of endogenous fecal and urinary excretions [5, 6]. The stable isotope method [7, 8], mineral-free diet [9], and mineral balance trials [5, 6] have been adopted to estimate endogenous excretions. Endogenous excretion estimated from intravenously injected mineral isotopes [7, 8] is probably the most accurately obtained, but it is an invasive, expensive, and labor-intensive approach. Mineral-free trials [9] may underestimate the endogenous excretion of animals at the production feeding level [10, 11]. Among these, mineral balance trials offer an approach to estimate endogenous excretion by analyzing the intercept of the regression of mineral excretion against intake [1, 5].

The National Academic of Science, Engineering, and Medicine (NASEM) committee of Dairy Cattle Nutrition [12] proposed a net requirement for maintenance of P calculated as the sum of endogenous fecal and urinary excretions. The endogenous fecal excretion of 1 g P/kg dry matter intake (DMI) was proposed from treatment means of three P balance trials with lactating dairy cows [13–15], and it was calculated assuming a true absorption coefficient of 0.80 [12]. The endogenous urinary excretion of 0.0006 g P/kg body weight (BW) was proposed from analysis of treatment means in three studies with lactating dairy cows [12]. The proposed net requirement for maintenance of Ca (0.90 g Ca/kg DMI) by NASEM 2021 [12] was derived from a regression between endogenous fecal excretion and DMI from treatment means of five studies where endogenous fecal excretion was obtained using intravenously injected radioisotopes of Ca [7, 8, 16–18], but only one involving lactating dairy cows [8].

Therefore, because of limited dataset used to derive NASEM 2021 models of P and Ca requirements [12], a more comprehensive factorial system to predict P and Ca requirements for lactating dairy cows requires development. Given the abundance of published P and Ca balance trials, we hypothesize that a new model derived from meta-regression of mineral balance trials may provide a robust estimate of endogenous fecal and urinary excretions, and MC of diet for P and Ca in diet.

Our primary objective was to derive a new system for P and Ca requirements for lactating dairy cows and to compare it with existing models of nutrient requirements of dairy cattle, such as National Research Council–NRC (2001) [19] and NASEM (2021) [12]. Specifically, we proposed derived new values for 1) endogenous fecal and urinary excretions (net requirements for maintenance) of P and Ca for lactating dairy cows from meta-regression analysis of mineral balance trials; 2) MC of diet for P and Ca; and 3) milk composition of P and Ca to predict the net requirement for lactation. The proposed P and Ca requirement system will be used as an updated mineral submodel of the NS Dairy Cattle (The Nutrition System for Dairy Cattle; [20]).

## Material and methods

### Dataset

A systematic review of mineral balance trials published as peer-review articles was performed to build our dataset. Treatment means were used to develop models to estimate P and Ca net requirements for maintenance and lactation, and MC diet for dairy cows. A first systematic review was performed in November 6, 2017 using the terms "dairy cows" and "phosphorus" in the Web of Science and Science Direct databases. A second systematic review was done in March 13, 2024 using the same terms, but included published $\geq$ 2018 year. A total of 349 peer-reviewed articles were initially found in first review and 78 articles from second review. The studies were selected based on the following criteria: (1) studies conducted with lactating dairy cows; (2) peer-reviewed articles; (3) reported treatment means of P or Ca intake, fecal and urinary excretions, and milk secretions; and (4) reported the standard error of the mean (SEM) or standard error of the difference (SED). When SED was reported in studies analyzed as a fixed model, SEM was calculated as SEM = SED /$\sqrt{2}$. A PRISMA flow chart showing the process of identification, exclusion, and inclusion of peer-reviewed articles to construct the P and Ca requirement model is described in Fig 1.

Based on these inclusion criteria, we selected 57 peer-review articles (first systematic review = 53, and second systematic review = 4) to data extraction (76 mineral balance trials (studies); total n = 298 treatment means; Fig 1; Table 1). No procedure to estimate missing data was adopted, except for the SEM of mineral fecal excretion. Data not reported in articles were evaluated as missing data; then, they were not used in final models. The complete dataset in the Excel® file is available in an open research data repository [21], and references used to develop models are available in S1 File.

### Dataset weighting

Each observation (treatment mean) was weighted by normalized inverse of the SEM [22] of mineral fecal excretion (g/d) as follows: Weighting factor = $W_1/W_2$, where: weighting factor = normalized inverse of the SEM of mineral fecal excretion (g/d); $W_1$ = 1/SEM mineral fecal excretion (g/d); and $W_2$ = overall mean of $W_1$ across studies. To prevent overweighting of studies with extremely low SEM [23], we truncated (i.e., trimmed) the SEM in 0.35 × overall mean SEM; then SEM < (0.35 × overall mean SEM) was trimmed at 0.35 × overall mean SEM of mineral fecal excretion. This analysis was conducted separately for the studies that adopted mixed and fixed effects models because mixed models tend to have higher SEM [22, 24]. Missing data of SEM of mineral fecal excretion were estimated using observed overall SEM across studies [25].

### Net requirement for maintenance

We assumed the net requirement for maintenance as the sum of the endogenous fecal and urinary mineral excretions. The endogenous fecal and urinary excretions of Ca and P were derived as the intercept of the regression between mineral intake (g/kg $BW^{0.75}$; predictor variable) and the sum of total fecal and urinary excretion (g/kg $BW^{0.75}$; response variable), using nonlinear meta-regression and adaptive Gaussian quadrature as the integration method, as follows:

$$Y_{ij} = \beta_1 \times e^{(\text{mineral intake} \times \beta_2)} + \text{trial}_j + e_{ij}, \tag{1}$$

where: $Y_{ij}$ = sum of total fecal and urinary excretion (g/kg $BW^{0.75}$) of the treatment means i of the mineral balance trial j; $\beta 1$ = overall intercept across all studies (fixed effects) and represents

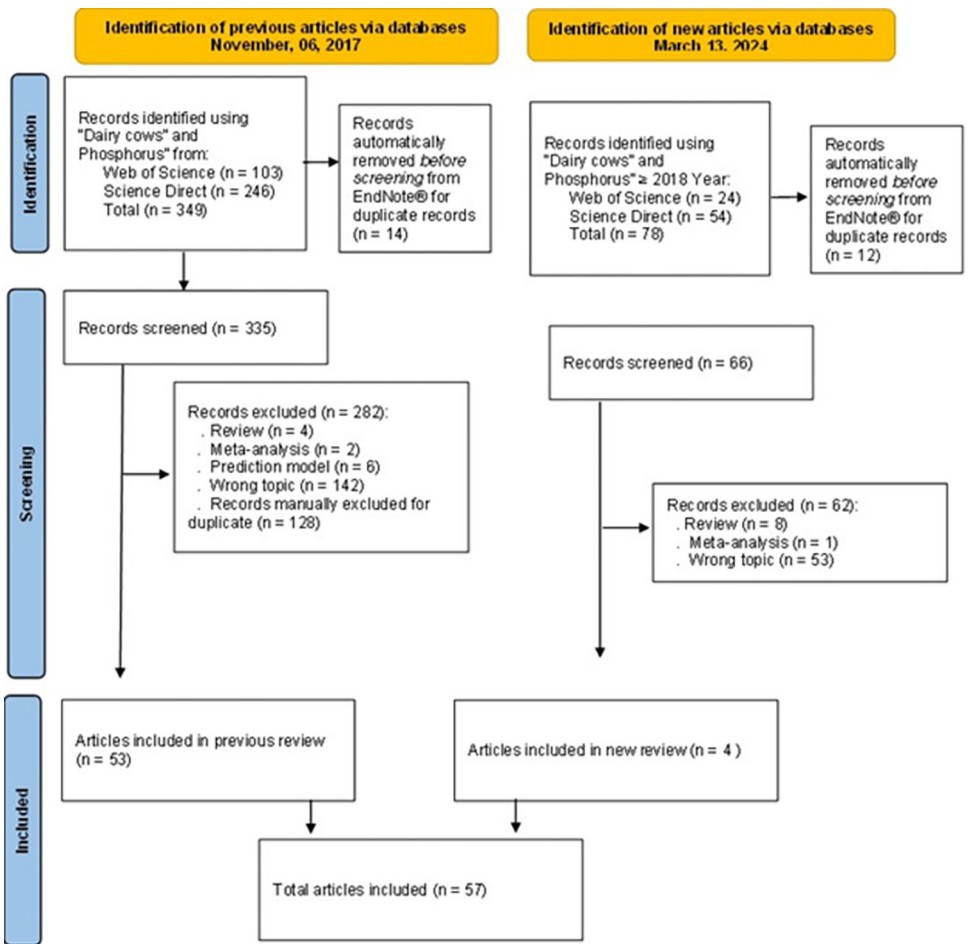

**Fig 1. PRISMA flow chart showing exclusion and inclusion criteria for selection of the peer-reviewed articles used to derive the phosphorus and calcium requirements system for lactating dairy cows.** A first systematic review was performed in November 6, 2017 (53 articles). The second systematic review was done in March 13, 2024 using the same terms, but included published $\geq$ 2018 year (n = 4 articles). A total of 57 peer-reviewed articles were included to create our complete dataset [21].

the sum of the endogenous fecal and urinary mineral excretion (g mineral/kg BW$^{0.75}$);
$\beta2$ = overall nonlinear statistics across all trials (fixed effect), without nutritional significance; trial$_j$ = random effect of mineral balance trial; and $e_{ij}$ = random error associated with each observation assuming a normal distribution (0, $\sigma^2$). The coefficients of the nonlinear models were initially defined from an iterative approach using graphic analysis. Observations were removed if the studentized residual was outside the range of -2.0 to +2.0 [21].

Random effect of study, and interaction between study × mineral intake on the sum of mineral fecal and urinary excretion were evaluated by mixed model analysis with variance component structure. The root square of estimated variance due to study (expressed as % mean dependent variable, [26]) was used as a heterogeneity index. We adopted values of square root of estimated variance due to study < 25%, 25–50% and > 50% as indicators of low, moderate, and high between-study heterogeneity [27]. Initially we evaluated the interaction effect of milk yield group ($\leq$ 20 kg/d; 20–30 kg/d; 30–40 kg/d; $\geq$ 40 kg/d) on intercept and slope between mineral intake and fecal + urinary output (Eq 1). If interaction between milk yield group and intercept or slope was not significant, an overall nonlinear equation was proposed.

**Table 1. Descriptive statistics of the complete dataset use to develop the new phosphorus and calcium requirement model for lactating dairy cows.**

| Item[1] | Mean | Median | Maximum | Minimum | SD | n[4] |
|---|---|---|---|---|---|---|
| Animal | | | | | | |
| BW (kg) | 564 | 590 | 754 | 316 | 100 | 237 |
| Milk yield (kg/d) | 28.3 | 30.8 | 52.8 | 4.50 | 11.0 | 255 |
| Days in milk | 115 | 106 | 367 | 13 | 69 | 226 |
| Milk protein (g/kg) | 30.8 | 31.1 | 41.5 | 3.2 | 5.9 | 148 |
| Milk fat (g/kg) | 38.6 | 37.1 | 53.8 | 14.7 | 6.03 | 146 |
| Milk lactose (g/kg) | 47.9 | 48.2 | 50.1 | 44.7 | 1.31 | 64 |
| Milk urea nitrogen (mg/dL) | 12.7 | 12.0 | 23.5 | 7.00 | 3.60 | 36 |
| Dry matter (DM) intake (kg/d) | 19.2 | 20.5 | 29.0 | 8.70 | 4.80 | 236 |
| Diet composition | | | | | | |
| Forage (g/kg DM) | 586 | 552 | 1000 | 258 | 155 | 254 |
| DM (g/kg as feed) | 519 | 512 | 923 | 398 | 105 | 64 |
| CP (g/kg DM) | 168 | 168 | 258 | 121 | 17.9 | 196 |
| EE (g/kg DM) | 32.3 | 33.1 | 47.0 | 17.4 | 8.18 | 64 |
| NDF (g/kg DM) | 338 | 341 | 496 | 260 | 49.0 | 190 |
| $NE_L$ (Mcal/kg DM) | 1.62 | 1.63 | 1.69 | 1.41 | 0.06 | 49 |
| P (g/kg DM) | 3.90 | 3.80 | 6.70 | 1.54 | 0.93 | 286 |
| Ca (g/kg DM) | 8.71 | 7.69 | 22.5 | 4.00 | 3.42 | 235 |
| P balance | | | | | | |
| P milk concentration (g/kg) | 0.91 | 0.92 | 1.27 | 0.53 | 0.13 | 209 |
| P intake (g/d) | 73.7 | 72.8 | 180 | 21.5 | 27.7 | 298 |
| P intake $(g/BW^{0.75}/d)^2$ | 0.62 | 0.58 | 1.48 | 0.18 | 0.22 | 237 |
| P fecal (g/d) | 43.2 | 40.4 | 118.8 | 10.9 | 20.5 | 274 |
| SEM P fecal (g/d) | 2.78 | 2.55 | 9.25 | 0.15 | 1.57 | 261 |
| P urinary (g/d) | 0.80 | 0.47 | 6.08 | 0.02 | 1.05 | 180 |
| P fecal + urinary $(g/BW^{0.75}/d)^3$ | 0.37 | 0.34 | 1.00 | 0.11 | 0.18 | 155 |
| P fecal/total excretion (g/g) | 0.98 | 0.99 | 0.99 | 0.89 | 0.02 | 155 |
| P milk (g/d) | 23.9 | 24.2 | 50.3 | 1.65 | 10.2 | 235 |
| Ca balance | | | | | | |
| Ca milk concentration (g/kg) | 1.25 | 1.20 | 2.24 | 0.86 | 0.28 | 113 |
| Ca intake (g/d) | 142 | 137 | 360 | 46.6 | 59.9 | 120 |
| Ca intake $(g/BW^{0.75}/d)^B$ | 1.45 | 1.34 | 3.68 | 0.38 | 0.69 | 120 |
| Ca fecal (g/d) | 94.0 | 89.3 | 212 | 21.2 | 42.8 | 120 |
| SEM Cal fecal (g/d) | 8.28 | 6.63 | 34.5 | 1.00 | 6.60 | 106 |
| Ca urinary (g/d) | 2.07 | 1.46 | 8.60 | 0.06 | 1.90 | 108 |
| Ca fecal + urinary $(g/BW^{0.75}/d)^C$ | 0.98 | 0.88 | 2.57 | 0.22 | 0.51 | 108 |
| Ca fecal/total excretion (g/g) | 0.97 | 0.98 | 0.999 | 0.76 | 0.05 | 108 |
| Ca milk (g/d) | 26.4 | 21.5 | 64.9 | 5.30 | 14.7 | 108 |

[1]Ca = calcium; CP = crude protein; DM = dry matter; EE = ether extract; NDF = neutral detergent fiber; $NE_L$ = Net energy for lactation; P = phosphorus; SEM = standard error of means.

[2]Calculated from each treatment mean as follows: mineral intake $(g/d)/BW^{0.75}$ (kg).

[3]Calculated from each treatment mean as follows: (mineral fecal excretion (g/d) + mineral urinary excretion $(g/d))/BW^{0.75}$ (kg). Some studies did not report the complete data of BW, mineral intake, fecal and urinary excretions; therefore, these studies (means treatment) automatically were not used on final model during the statistical analysis.

[4]Treatment means of 76 balance trials (studies) in 57 peer-review articles (list of reference is available in S1 File). The complete dataset is available in an Excel® file from [21]. Treatment means removed as outliers in finals models: P fecal and urinary excretions (n = 27); Ca fecal and urinary excretions (n = 17); P-MC (n = 8); P milk content (n = 6). A list of observations removed from analysis of studentized residual (outside the range of -2.0 to +2.0) is also available in [21].

Observations were removed if the studentized residual was outside the range of -2.0 to +2.0. The list of removed observations (outliers) on final models is available in [21]. Significance was declared at $P \leq 0.05$. Analyses were conducted using the PROC MIXED and PROC NLMIXED procedures [28] of the SAS® On Demand for Academics Analyses. As the WEIGHT statement is not available on PROC NLMIXED procedure, the REPLICATE statement was adopted as a WEIGHT statement when the PROC NLMIXED procedure was used [29]. The dataset used to derivate the net requirement for maintenance models included only studies from the first systematic review. The final dataset, codes and outputs are described in S2.1 and S2.2 in S2 File.

## Metabolizable coefficient

The metabolizable coefficient of phosphorus and calcium of each observation (treatment means) was quantified as follows:

$$MC\ (0\ to\ 1) = \frac{intake - fecal\ excretion - urinary\ excretion + fecal\ and\ urinary\ endogenous\ exection}{intake}, \quad (2)$$

where: intake (g/d) = observed P or Ca intake reported from studies; fecal excretions (g/d) = observed P or Ca total fecal excretion reported from studies; urinary excretion (g/d) = observed P or Ca total urinary excretion reported from studies; and the sum of fecal and urinary endogenous excretions of P and Ca were estimated from Eqs 3 and 4, respectively (Table 2).

To identify potential dietary factors affecting MC, we initially analyzed the interaction effect of dietary characteristics (forage in diet, neutral detergent fiber (NDF), crude protein (CP), ether extract, and starch) with MC using a bivariable mixed model with unstructured variance and considering the balance trial as a random effect [22]. The root square of estimated variance due to study also was used as a heterogeneity index for proposed MC model as early informed. Observations were removed if the studentized residual was outside the range of -2.0 to +2.0 [21]. Significance was declared at $P \leq 0.05$. Analyses were conducted using PROC MIXED [22] of the SAS® On Demand for Academics. The complete dataset used to derivate MC values included studies of first and second systematic review. Final dataset, codes and outputs are described in S2.3 in S2 File.

**Table 2. Nonlinear mixed regression analysis of sum of total fecal and urinary excretions of phosphorous ($P_{FU}$) or calcium ($Ca_{FU}$) and mineral intake to obtain the net requirements for maintenance of lactating dairy cows.**

| Equation number | Equation[1] | Cross validation 5-fold[2] | | | | n[3] | Net requirements for maintenance[4] (g/d) |
|---|---|---|---|---|---|---|---|
| | | r | $C_b$ | CCC | RMSPE (% observed) | | |
| 3 | $P_{FU}$ (g/BW$^{0.75}$/d) = 0.1352$_{\pm0.0427}$ (P = 0.004) $\times e^{(1.4010\pm0.1863\ (P < 0.001)\ \times\ P\ Intake\ (g/BW0.75/d))}$ | 0.89 | 0.96 | 0.85 | 24.1 | 130 | P = 0.1352$_{\pm0.0427}$ $\times$ BW$^{0.75}$ |
| 4 | $Ca_{FU}$ (g/BW$^{0.75}$/d) = 0.3604$_{\pm0.1438}$ (P = 0.0251) $\times e^{(0.5925\pm0.1306\ (P < 0.001)\ \times\ Ca\ Intake\ (g/BW0.75/d))}$ | 0.79 | 0.97 | 0.77 | 20.5 | 70 | Ca = 0.3604$_{\pm0.1438}$ $\times$ BW$^{0.75}$ |

[1] BW = body weight; Ca = calcium; P = phosphorous. No interaction effect of milk yield group on intercept ($P = 0.886$) and slope ($P = 0.886$) were observed for P fecal and urinary excretion; and also no interaction effect of milk yield group on intercept ($P = 0.918$) and slope ($P = 0.899$) were observed for Ca fecal and urinary excretion.

[2] CCC = concordance correlation coefficient; r = correlation coefficient (precision); $C_b$ = bias correction factor (accuracy); and RMSPE = root mean square prediction error. The five folds were previously created by study grouping to guarantee independence between observed and predicted values (Fig 3). Cross validation 5-fold procedure, codes and outputs are described in S2.1 and S2.2 in S2 File.

[3] Treatment means reported from 39 (P) and 23 (Ca) balance trials used on final models after removed from analysis of studentized residual (Table 1)

[4] Estimated intercept between $P_{FU}$ or $Ca_{FU}$ and mineral intake (fecal + urinary endogenous excretion; Fig 2).

## Model evaluation

The proposed models to predict mineral total excretion (fecal + urinary) were evaluated by linear regression between observed (dependent variable) and predicted (independent variable) values using the 5-fold cross-validation approach [30]. The five folds for each model were previously created by study (mineral balance trial) grouping to guarantee independence between observed and predicted values (external model evaluation). The slope and intercept between observed and predicted MP values were tested to quantify the magnitude of the mean bias and linear bias of models, respectively. Estimates of correlation coefficient (r; precision), bias correction factor ($C_b$; accuracy), coefficient of concordance correlation (CCC; combined precision and accuracy), and root mean square prediction error (RMSPE; accuracy) were obtained using the *metrica* Package [31] of the R Software, version 4.3.1. Milk mineral concentration between breeds cows was compared from 95% confidence interval (95% CI) analysis. The dataset used to evaluate the net requirement for maintenance models included only studies of the first systematic review. The dataset, the 5-fold cross-validation procedure, codes and outputs of statistics analysis are described in S2.1 and S2.2 in S2 File.

## Results

### Dataset

Our complete dataset built to develop our P and Ca requirement system comprised data from 11 countries and represented a wide range of lactating dairy cows performance (milk yield of 4.5 to 52.8 kg/d; BW of 316 to 754 kg; dry matter intake of 8.7 to 29.0 kg/d; 13 to 367 days in milk) and dietary characteristics (258 to 1000 g forage/kg dry matter (DM) diet; 260 to 496 g NDF/kg DM; 1.54 to 6.7 g P/kg DM; 4 to 22.5 g Ca/kg DM) (Table 1). Multiparous cow datasets were reported in 31 studies, primiparous in three studies, multiparous and primiparous cows in eight studies. Parity was not was reported in 34 studies. The United States was the primary country of origin for the studies (67.4%), followed by Canada (7.4%), the UK (6.7%), Sweden (4.7%), and Germany (4.0%). Holstein was the predominant breed (67.9%), followed by Jersey (17.2%). Continuous trials were the most frequently adopted experimental design (67.1%) and TMR was the predominant feeding system (93.7% observations). Fecal and urinary P and Ca excretions were obtained by total collection in 75.6% of observations; other studies (24.4% observations) used ytterbium (6.5% observations), $Cr_2O_3$ (5.5%), indigestible NDF or acid detergent fiber 288 h (3.6%), n-alkanes (3.6%), and lignin, $TiO_2$ and other (5.2%) as fecal output markers, and creatinine urine as urinary output marker.

Fecal excretion was the primary pathway excretion of P (98% total excretion) and Ca (97%) (Table 1).

### Net requirement for maintenance

No interaction effect of milk yield group on intercept ($P = 0.886$) and slope ($P = 0.886$) were observed for P fecal and urinary excretion; and also no interaction effect of milk yield group on intercept ($P = 0.918$) and slope ($P = 0.899$) were observed for Ca fecal and urinary excretion (Table 2). Therefore, overall nonlinear equations were used to estimate endogenous excretion of P and Ca.

The estimated net requirement for maintenance for P (g/d) = $0.1352_{\pm 0.0427} \times BW^{0.75}$, and Ca (g/d) = $0.3604_{\pm 0.1438} \times BW^{0.75}$ (Table 2). The nonlinear mixed models of P (Eq 3) and Ca excretions (Eq 4) to obtain the endogenous excretions (net requirement for maintenance) had a suitable precision (r = 0.89 and 0.79) and accuracy ($C_b$ = 0.96 and 0.97; CCC = 0.85 and 0.77; and RMSPE = 24.1 and 20.5% observed) (Table 2, Figs 2 and 3). No adjustment of

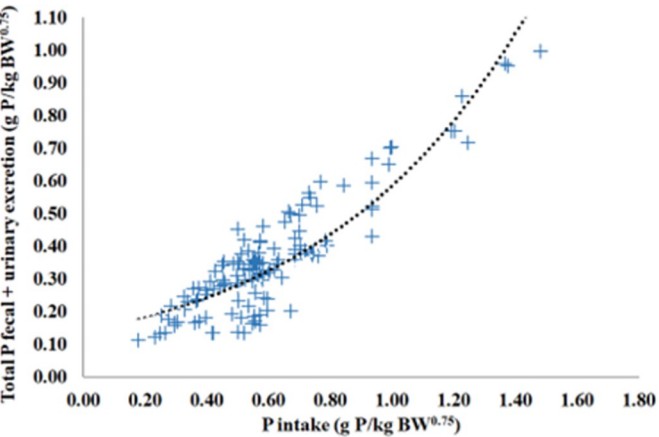

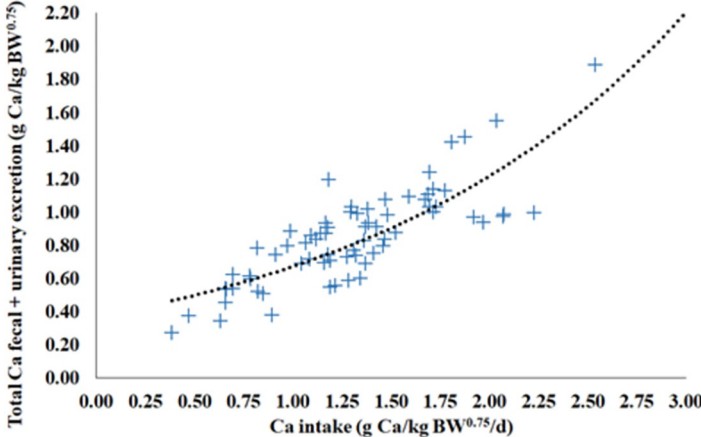

**Fig 2. Relationship between the sum of daily phosphorus (P) or calcium (Ca) total fecal and urinary excretion and mineral intake.** The dotted lines are the predicted values from Eqs 3 and 4 (Table 2). n = 130 treatment means reported from 39 P balance trials, and n = 70 treatment means reported from 23 trials Ca balance trials.

maintenance requirement for genotype was proposed due to the predominance of Holstein breed data in our mineral balance dataset. No evidence of mean biases (intercept $\neq$ zero; $P \geq 0.10$) and linear bias (slope $\neq 1$; $P \geq 0.10$) for P and Ca was observed (Fig 3).

The sum of P total fecal and urinary excretions was affected by random study ($P < 0.01$), but no effect of study × P intake ($P = 0.21$) was observed (S3.1 in S3 File). The root squared of study variance (a proxy for between-study heterogeneity) represented 24.8% mean of P total excretion (S3.1 in S3 File). The sum of Ca total fecal and urinary excretions was affected by random study ($P < 0.01$) and study × Ca intake ($P < 0.01$), and the root squared of study variance represented 30.2% mean of P total excretion (S3.2 in S3 File).

## Metabolizable coefficient and mineral milk concentration

Dietary forage level, CP, ether extract, NDF and starch did not affect MC-P and MC-Ca (Table 3). Therefore, an overall mean (± standard error) estimated diet MC-P = $0.69_{\pm 0.01}$ and MC-Ca = $0.65_{\pm 0.02}$ were proposed (Fig 4).

Mineral (P and Ca) milk concentration of Holsteins was lower ($P \leq 0.05$) than Jersey cows (Fig 5). The mean P milk concentration was 0.90 (95% CI; 0.89, 0.92) g/kg for Holstein and

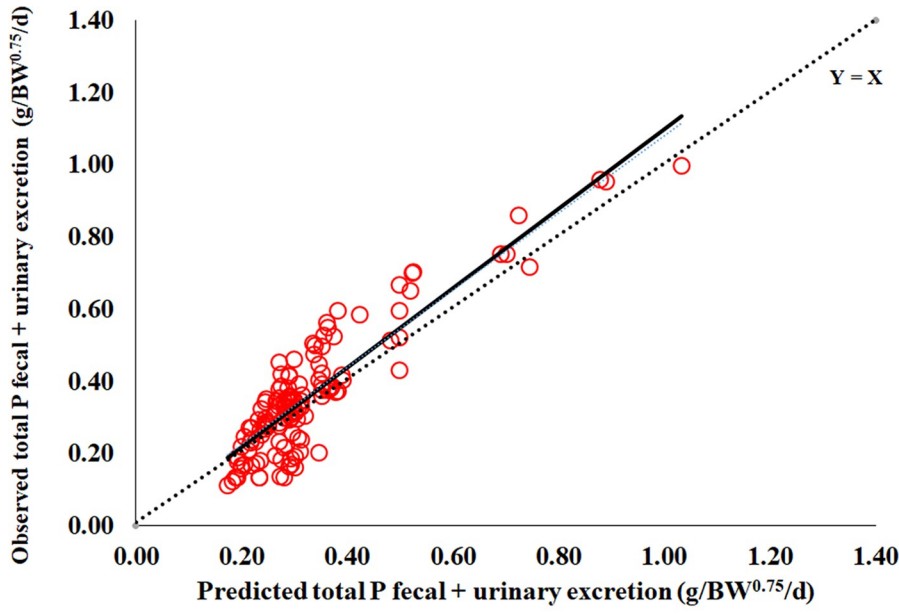

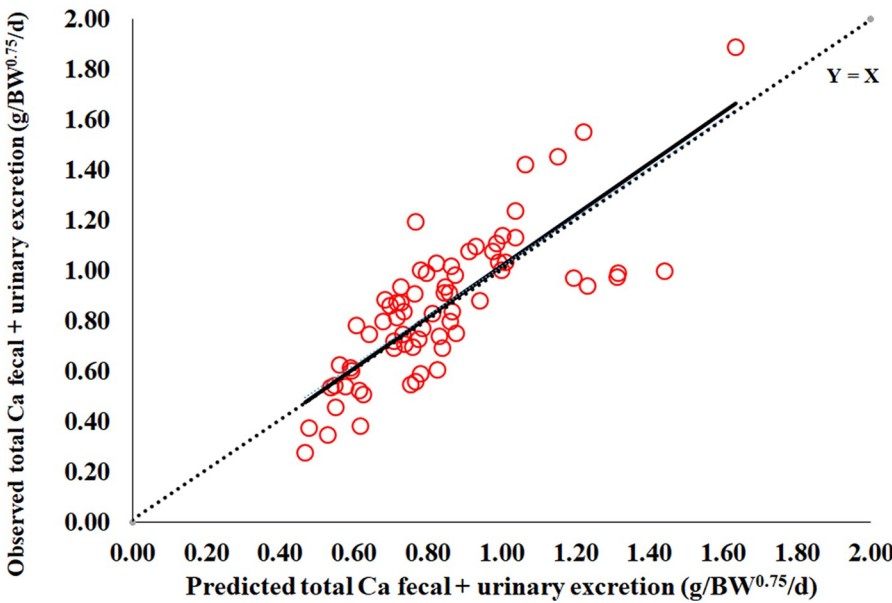

**Fig 3. Plot of observed *versus* predicted total phosphorus ($P_{FU}$) or calcium ($Ca_{FU}$) fecal and urinary excretion (prediction equations are in Table 2).** Predicted values were derived from the 5-fold cross-validation procedure. The five folds were previously created by study grouping to guarantee independence between observed and predicted values. n = 130 treatment means reported from 39 P balance trials, and n = 70 treatment means reported from 23 trials Ca balance trials.

1.00 (95% CI; 0.96, 1.04) g/kg for Jersey, while Ca milk concentration was 1.18 (95% CI; 1.12, 1.23) g/kg for Holstein and 1.38 (95% CI; 1.28, 1.47) g/kg for Jersey (Fig 5). These values were used to quantify the net requirement for lactation in our model (Table 4). A summary of our

**Table 3. Effects of the diet composition on estimated diet metabolizable coefficient of phosphorous (MC-P) and calcium (MC-Ca) in lactating dairy cows.**

| Diet composition[3] | MC-P[1] P-value (n)[2] | MC-Ca[1] P-value (n)[2] |
|---|---|---|
| Forage in diet (g/kg DM) | 0.514 (129) | 0.479 (70) |
| Crude protein (g/kg DM) | 0.392 (84) | 0.267 (31) |
| Ether extract (g/kg DM) | 0.154 (35) | - |
| Neutral detergent fiber (g/kg DM) | 0.663 (83) | 0.797 (31) |
| Starch (g/kg DM) | 0.837 (26) | - |

[1]MC-P and MC-Ca were obtained from Eq 2 (in text).

[2]Each variable was evaluated from a bivariable mixed model with variance components and weighted by inverse on normalized SEM mineral fecal excretion; n = treatment means reported trials balance (Table 1). Final dataset, codes and outputs are described in S2.3 in S2 File.

[3] DM = dry matter.

proposed system of net requirements for maintenance and lactation, and dietary requirements for P and Ca is shown in Table 4. The dietary requirement is the sum of the net requirements for maintenance and lactation divided by the dietary MC.

## Discussion

Our primary objective was to establish a new factorial P and Ca requirements system for maintenance and lactation from a meta-regression of a comprehensive mineral balance trials database. Our dataset represented a wide range of animal performance, including dairy cows with low to very high milk yield (4.5 to 52.8 kg/d; BW of 316 to 754 kg), which is aligned with our objective of deriving a comprehensive requirement system. It is noteworthy that we addressed

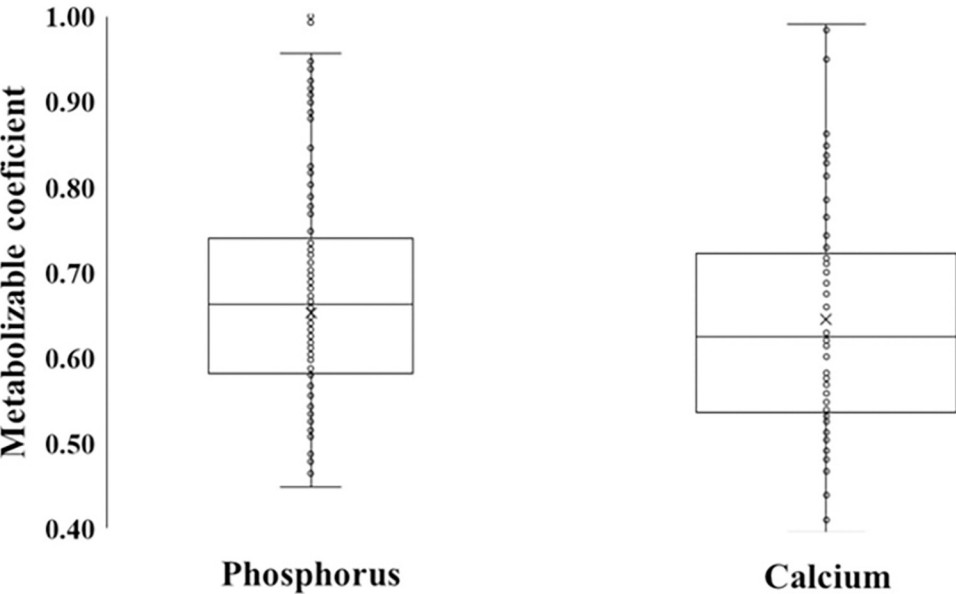

**Fig 4. Box plot of the estimated diet metabolizable coefficient of phosphorus and calcium for lactating dairy cows from Eq 2.** Estimated metabolizable coefficient for phosphorus: mean ± standard error (SE) = $0.69_{\pm0.01}$ and n = 157 treatment means. Estimated metabolizable coefficient for calcium: mean ± SE = $0.65_{\pm0.02}$ and n = 81 treatment means. Final dataset, codes and outputs are described in S2.1 and S2.2 in S2 File.

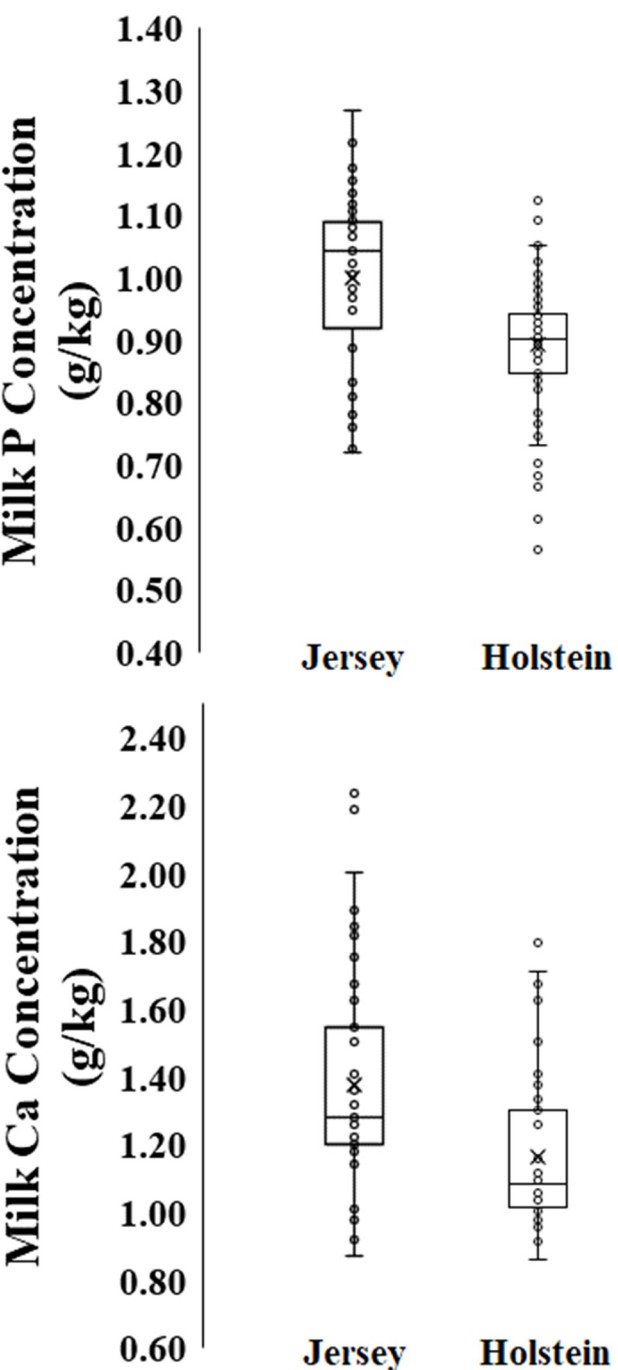

**Fig 5. Box plot of milk phosphorus (P) and calcium (Ca) concentration for Holstein and Jersey cows (complete dataset, Table 1).** Milk P mean (95% confidence interval): 0.90 (0.89, 0.92) g/kg milk (n = 135 treatment means) for Holstein; and 1.00 (0.96, 1.04) g/kg milk for Jersey (n = 41). Milk Ca mean (95% confidence interval): 1.18 (1.12, 1.23) g/kg milk (n = 56 treatment means) for Holstein; and 1.38 (1.28, 1.47) g/kg milk for Jersey (n = 47).

potential outliers arising from factors such as milk yield, mineral intake and/or excretion ensuring their remotion from analysis based on studentized residual. Furthermore, the suitable precision and accuracy, and the absence of significant prediction biases in models indicates that the estimation of the endogenous fecal and urinary excretion (net mineral requirement

**Table 4. Summary of proposed factorial system of phosphorus and calcium requirements for maintenance and lactation of dairy cows.**

| Item | Phosphorus[1] (mean ± SE) | Calcium[1] (mean ± SE) |
|---|---|---|
| Net requirement for maintenance ($NR_M$, g/d) | $0.1352_{\pm 0.0427} \times BW^{0.75}$ | $0.3604_{\pm 0.1438} \times BW^{0.75}$ |
| Net requirement for lactation ($NR_L$, g/d) | Holstein = $0.90_{\pm 0.01} \times MY$<br>Jersey = $1.00_{\pm 0.02} \times MY$ | Holstein = $1.18_{\pm 0.03} \times MY$<br>Jersey = $1.38_{\pm 0.05} \times MY$ |
| Metabolizable coefficient (MC)[2] | $0.69_{\pm 0.01}$ | $0.65_{\pm 0.02}$ |
| Dietary intake requirement (DIR, g/d) | $(NR_M + NR_L)/MC$ | $(NR_M + NR_L)/MC$ |
| Dietary requirement (g/kg DM) | DIR/DMI | DIR/DMI |

[1]BW = body weight; DMI = dry matter intake (kg/d); MY = milk yield (kg/d); SE = standard error.

[2]Actual MC of diets should preferably be used if accurately know.

for maintenance) from intercept between excretion and mineral intake was unbiased [32, 33]. In addition, as we previously created the folds by mineral balance trial grouping to guarantee independence between model development (train) and evaluation (test), our models of mineral excretion were evaluated using an external model evaluation approach.

Although we observed random effect of study on P and Ca excretions, the root squared of study variance (a proxy for between-study heterogeneity) of the sum of fecal and urinary excretions of P (24.8% mean) and Ca (30.2% mean) can be considered low and moderate [27, 34]. These results indicate the effects of mineral intake on the sum of total fecal and urinary excretion were consistent across studies. Moreover, as our models were adjusted for the random effect of study, the between-study variance was captured in the final model.

Our non-linear model of mineral excretion (fecal + urinary) from mineral intake also allowed to capture homeostasis mechanisms involved in absorption of P and Ca. Diets with more than approximately 0.8 g P/kg $BW^{0.75}$ and 1.7 g Ca/kg $BW^{0.75}$ seemingly increase the rate of excretion of P and Ca, respectively (Fig 2). These changes may be a result of the animal downregulating the efficiency of transcellular intestinal mineral absorption when diets exceed the body mineral requirements [35]. Although our dataset also contains observations with high P and Ca intakes, we reiterate that: 1) discrepant observations were removed from model based on the analysis of studentized residuals, 2) the exponential model of P and Ca excretion from mineral intake exhibited low to moderate between-study variance (heterogeneity), and 3) most importantly, the models exhibited suitable precision and accuracy and no significant prediction biases were observed.

In this study, we introduced the term "metabolizable coefficient" for minerals, replacing the term "absorption coefficient" since MC was obtained from fecal and urinary excretion. The term "absorption coefficient" is more appropriate when derived solely from fecal excretion. The concept of mineral metabolizability aligns with the mineral retention coefficient [6]. Our study indicates that P and Ca urinary excretion accounted for less than 3% of the total excretion (Table 1), confirming previous findings that urinary excretion of Ca and P is quantitatively negligible in dairy cows [36–38]. Therefore, in practical terms, the MC and AC are quantitatively similar for lactating dairy cows.

The higher milk P and Ca concentrations of Jersey compared to Holstein in our study can be attributed to the higher milk solid content of Jersey cows, particularly the milk casein content [39, 40]. On average, 70% of Ca and 50% of inorganic phosphate are located in the casein micelle [39]. The milk Ca concentration value of our study is higher than that adopted by NASEM (2021, [12]) of 1.03 and 1.17 g/kg for Holstein and Jersey cows, but it is similar to obtained values in some herd-level studies [41–43] and is closer to NRC (2001, [19]). However,

when mineral in milk is feasibly measured in commercial herds, we suggested to use the actual P and Ca milk concentration to calculate the net requirement for lactation.

Our second objective was to compare the proposed model with the NASEM (2021) model [12]. The NRC (2001) model [19] also was compared because it has been adopted for predicting P and Ca requirement in several other dairy cattle nutrition models [44, 45]. Our model for Ca and P net and dietary requirements was developed using a different approach and a larger and more comprehensive dataset than that adopted by the NASEM (2021) committee [12]. We estimated the endogenous fecal and urinary (net requirements for maintenance) from the intercept of a nonlinear equation between mineral intake and the sum of total fecal and urinary excretions, using 130 means treatment from 39 balance trials of lactating dairy cows for the P model, and 70 means treatment from 23 balance trials for Ca the model.

The NASEM (2021) committee [12] also proposed accounting for the net requirement for maintenance of P as the sum of endogenous fecal and urinary excretions. The endogenous fecal excretion of 1 g P/kg DMI was proposed based on treatments mean from only three P balance trials with lactating dairy cows [13–15]. This value was calculated assuming a true absorption coefficient of 0.80 [12]. The endogenous urinary excretion of 0.0006 g P/kg BW was proposed based on the analysis of treatment means in three studies with lactating dairy cows [12]. The proposed net requirement for maintenance of Ca (0.90 g Ca/kg DMI) by NASEM (2021) was derived from a regression between endogenous fecal excretion and DMI of treatment means in five studies where the endogenous Ca fecal excretion was obtained by intravenously injected radioisotopes of Ca [7, 8, 16–18], but only one study involving lactating dairy cows [8]. Therefore, our proposed system of P and Ca requirements for lactating dairy cows is based on a different approach and a larger size scope of dataset than that adopted in NASEM (2021 [12].

Our model predicts net P requirements (maintenance plus lactation) 12% lower than the NASEM (2021) [12] and 4% lower than the NRC (2001) [19] recommendations for a 500 kg BW dairy cow producing 10 kg milk per day (Fig 6). For cows producing 50 kg milk per day (700 kg BW), our model predicts net P requirements 17% lower than the NASEM (2021) [12] and 14% lower than the NRC (2001) [19] recommendations. Similarly, predicted P dietary requirement of our model was 6% lower than the NASEM (2021) [12] and 8% lower than the NRC (2001) [19] recommendations for a 500 kg BW dairy cow producing 10 kg milk per day (Fig 6). For cows producing 50 kg milk per day (700 kg BW), our model predicts P dietary requirements 13% lower than the NASEM (2021) [12] and 16% lower than the NRC (2001) [19] recommendations (Fig 6).

As our proposed MC-P (0.69) is similar to overall absorption coefficient adopted by NASEM (2021; 0.72) and NRC (2001; 0.70 for concentrate and 0.64 for forage), the lower P net requirements (mainly maintenance) in our model explains the lower P dietary requirements. Phosphorus is the most expensive macromineral supplemented in dairy cattle diets, sourced from nonrenewable minerals. Excessive excretions from P overfeeding can contribute to soil and aquatic ecosystem eutrophication [3, 4]. Therefore, our proposed model may contribute to elaborate more profitable and environmentally sustainable diets for dairy cows, if our model does not result in P underfeeding.

Our model estimated total dietary requirements of 63 to 92 g P/cow/d for cows producing 30 to 50 kg milk/d (Fig 6). Therefore, assuming predicted DMI of 20.7 and 27.0 kg/cow/d (NASEM, 2021), our model estimates total dietary requirements of 3.0 to 3.4 g P/kg DM diet for cows producing 30 to 50 kg milk/d, respectively. Wu et al. (2000) [46] reported no effect on milk yield, reproductive performance and health records of lactating cows (overall lactation milk yield of 35 to 37 kg/d) fed diet with 3.1, 4.0 or 4.9 p P/kg DM. No effect on milk yield of cows producing about 35 kg milk/d fed diets with 3.3 or 4.2 g P/kg DM [47], or cows

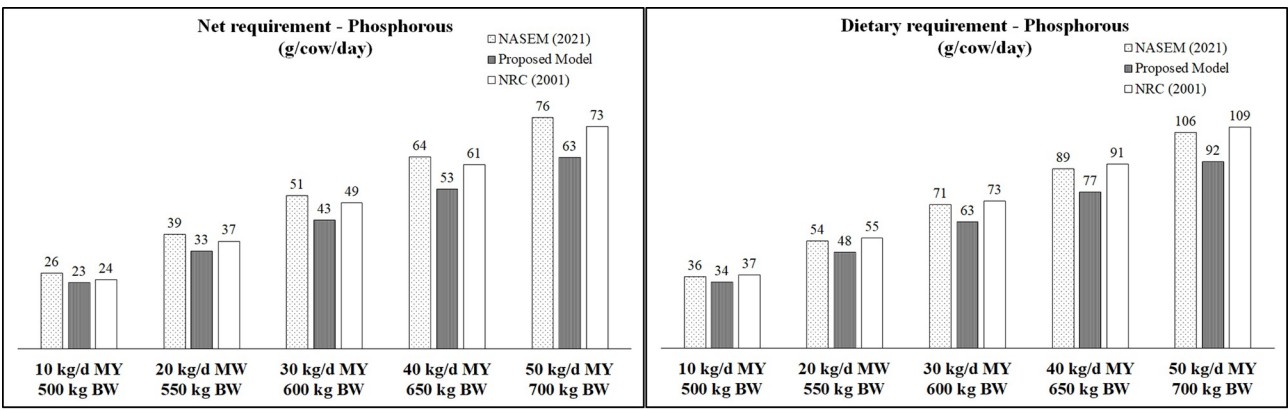

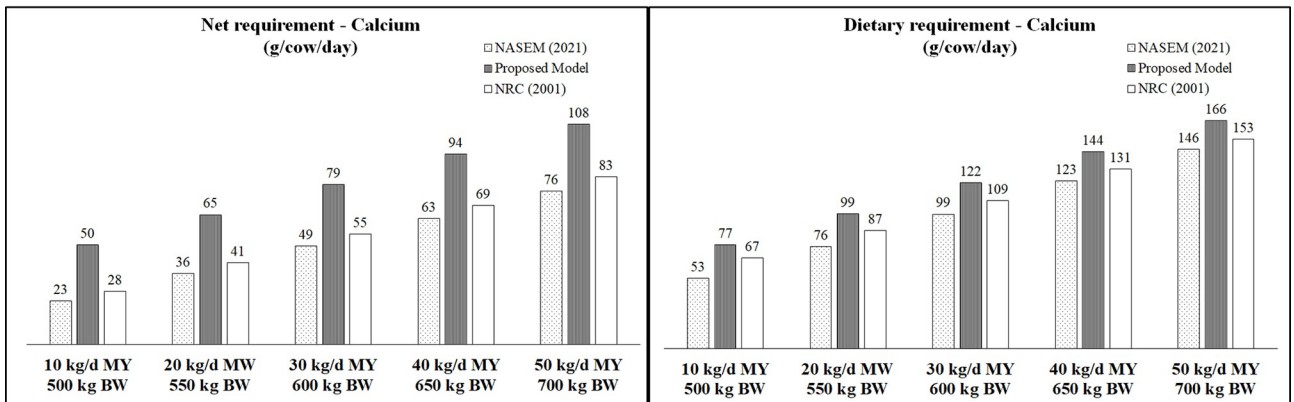

**Fig 6. Estimated net and dietary requirements of phosphorus and calcium for lactating dairy cows from our proposed system (Table 4), NASEM (2021, [12]) and NRC (2001, [19]) models.** NASEM (2021) P requirement: maintenance (g P /d) = 1 × dry matter intake (DMI; kg/d) + 0.0006 × body weight (BW; kg); lactation (g P/d) = 0.90 × milk yield (kg/d); overall absorption coefficient (default) = 0.72; estimated DMI for multiparous cows with 60 days in milk, three points of body condition score, and milk energy of 0.73 Mcal/kg. NASEM (2021) Ca requirement: maintenance (g Ca/d) = 0.9 × DMI (kg/d); lactation (g P/d) = 1.03 × milk yield (kg/d); absorption coefficient = 0.60 for concentrate and 0.40 for forage; assuming forage in diet of 800, 600, 500, 450 and 400 g/dry matter (DM) for 10, 20, 30, 40 and 50 kg/d of milk yield, respectively. NRC (2001) P requirement: maintenance (g P /d) = 1 × DMI (kg/d) + 0.0002 × BW (kg); lactation (g P/d) = 0.90 × milk yield (kg/d); absorption coefficient = 0.70 for concentrate and 0.64 for forage. NRC (2001) Ca requirement: maintenance (g Ca/d) = 0.031 × BW (kg); lactation (g Ca/d) = 1.22 × milk yield (kg/d); absorption coefficient = 0.60 for concentrate and 0.30 for forage; assuming forage in diet of 800, 600, 500, 450 and 400 g/DM for 10, 20, 30, 40 and 50 kg/d of milk yield, respectively.

producing 43 kg/d fed diets with 3.2 or 4.4 g P/kg DM [48] were also reported. A long term feeding trial (two lactations) of limited dietary P supply (3.3, 2.8 and 2.4 g P/kg DM diet) indicated that dietary P had no effect on reproductive performance, but intake and milk yield were reduced with 2.4 g P/kg DM, suggesting that the diets with 2.8 g P/kg DM was sufficient to meet the P requirement of dairy cows producing approximately 9000 kg of milk per lactation [49]. Keanthao et al. (2021) [50] reported that a reduction of dietary P from 3.8 to 2.9 g/kg during first eight weeks after calving improved plasma Ca levels without compromising diet intake and milk production (mean = 43.3 kg milk/d). Therefore, based on these limited number of dose response experiments, our model seems to adequately estimate P requirements for high production dairy cows.

In contrast with P model, our model predicts a net Ca requirement (maintenance plus lactation) 117% higher than the NASEM (2021) [12] and 79% higher than the NRC (2001) [19] recommendations for a 500 kg dairy cows producing 10 kg milk per day (Fig 6). For a cow

producing 50 kg milk per day (700 kg BW), our model predicts a net Ca requirements 42% higher than the NASEM (2021) [12] and 30% higher than the NRC (2001) [19] recommendations. However, due to the higher MC-Ca in proposed model than the absorption coefficient for Ca adopted by NASEM (2021) and NRC (2001), the differences in dietary Ca requirements between our model and NASEM (2021) and NRC (2001) recommendations were smaller than the net Ca requirements (Fig 6). Our model predicts a dietary Ca requirement 45% higher than the NASEM (2021) [12] and 15% higher than the NRC (2001) [19] recommendations for a 500 kg dairy cows producing 10 kg milk per day. For cows producing 50 kg milk per day (700 kg BW), our model predicts a dietary Ca requirement 14% higher than the NASEM (2021) [12] and 8% higher than the NRC (2001) [19] recommendations (Fig 6). Due to the positive relationship between MC and endogenous excretion (Eq 2), the higher value of MC in proposed model than the absorption coefficient for Ca adopted by NASEM (2021, [12]) and NRC (2001, [19]) may partially explain the higher Ca endogenous excretion in our model.

### Limitations

Our study has some limitations. First, the endogenous excretion (maintenance requirement) was obtained from a mathematical extrapolation for zero balance. By definition, zero balance represents the intake required to maintain an existing pool size and not necessarily "the requirement" for a mineral element [11]. Therefore, estimated endogenous excretion from balance trial represents an approximation of mineral requirement for maintenance, and it depends of the amount and bioavailability of the mineral under study [11].

Second, although fecal and urinary P and Ca excretions were obtained by total collection in most studies in our dataset, we also included studies that used external ($Cr_2O_3$, $TiO_2$, and Ytterbium) and internal (indigestible NDF or ADF, n-alkanes, and lignin) fecal markers output, and urine creatinine as urinary output marker from spot sampling. Although there is evidence that these external and internal fecal markers can accurately estimates fecal output [51–54], and that urine creatinine can be an accurate maker for volume and minerals urinary output [55, 56], the variance is potentially higher than total collection. Therefore, the use of treatment means of mineral excretion obtained from fecal and urinary markers can partially explain the between-study heterogeneity of ours models.

Third, although our nonlinear equation to estimate P and Ca endogenous excretion exhibited suitable precision and accuracy, no significant prediction biases, and low to moderate between-study variance (heterogeneity), other factors can are involved in endogenous fecal losses, as such mineral saliva secretion, rumen microbial mineral outflow, and DMI [36, 57]. Fourth, we proposed fixed values to predict dietary MC-P and MC-Ca. However, intestinal absorption of P and Ca may be affected by source, mineral antagonism, physiology stage, 1,25-dihydroxy vitamin D status, and mineral homeostasis [12, 35, 58]. Therefore, when actual MC feeds or diets are accurately known, they should be used to replace our proposed true MC for predicting dietary requirements. Finally, although the proposed equations to estimate endogenous excretions were independently evaluated from a 5-fold cross-validation approach [30], the adequacy of our proposed system for predicting dietary P and Ca requirements for dairy cows and other models, as well NRC (2001) [19] and NASEM (2021) [12] models still needs to be further evaluated through independent response-dose feeding experiments.

### Conclusions

We have established a new factorial system for accounting net and dietary P and Ca requirements for maintenance and lactation based on a meta-regression of mineral trials involving lactating dairy cows. The estimation of endogenous fecal and urinary (net requirements for

maintenance) was derived from intercept of a nonlinear equation between mineral intake and the sum of total fecal and urinary excretions. Our proposed model provided a suitable precision and accuracy for predicting endogenous fecal and urinary excretions through of a 5-fold cross-validation analysis. An overall metabolizable coefficient of dietary P and Ca were proposed.

Our new system estimates lower net and dietary requirements of P for lactation dairy cows compared to the NASEM (2021) and NRC (2001) models, but higher Ca requirement than NASEM (2021) and NRC (2001). Therefore, our P model may contribute to elaborate more profitable and environmentally sustainable diets for dairy cows. However, the adequacy of our proposed system predicting dietary P and Ca requirements and other models, such as the NASEM (2021) and NRC (2001), still requires further evaluation through independent response-dose feeding experiments. In addition, our model can likely be improved through future studies that evaluate dietary, animal and environmental variables affecting mineral endogenous excretions and metabolizable coefficients.

## Supporting information

**S1 File. Publications used to development the phosphorus and calcium requirement system for lactating dairy cows.**
(DOCX)

**S2 File. Final dataset, codes used to derive the P excretion nonlinear model, quantify study variance, and cross-validation procedure (S2.1), Final dataset, codes used to derive the Ca excretion nonlinear model, quantify study variance, and cross-validation procedure (S2.2), and Final dataset, and codes used to derive Phosphorous and Calcium Metabolizable Coefficient (S2.3).**
(DOCX)

**S3 File. Relationship between the sum of daily phosphorus (P) total fecal and urinary excretion and P intake (S3.1), and Relationship between the sum of daily phosphorus (Ca) total fecal and urinary excretion and Ca intake (S3.2).**
(DOCX)

**S4 File. The PRIMA 2020 checklist.**
(DOCX)

## Author Contributions

**Data curation:** André Soares de Oliveira.

**Formal analysis:** André Soares de Oliveira.

**Funding acquisition:** André Soares de Oliveira.

**Investigation:** André Soares de Oliveira, Suziane Rodrigues Soares.

**Methodology:** André Soares de Oliveira.

**Project administration:** André Soares de Oliveira.

**Resources:** André Soares de Oliveira.

**Supervision:** André Soares de Oliveira.

**Validation:** André Soares de Oliveira.

**Visualization:** André Soares de Oliveira.

**Writing – original draft:** André Soares de Oliveira.

**Writing – review & editing:** André Soares de Oliveira.

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
