## [Decision Letter · Decision Letter 0]

25 Apr 2024

PONE-D-24-09154A new system of phosphorus and calcium requirements for dairy cowsPLOS ONE

Dear Dr. Oliveira,

Thank you for submitting your manuscript to PLOS ONE. After careful consideration, we feel that it has merit but does not fully meet PLOS ONE’s publication criteria as it currently stands. Therefore, we invite you to submit a revised version of the manuscript that addresses the points raised during the review process. Dear Authors, 

Reviewers has reviewed the manuscript and all of the 3 reviewers suggested minor revision. Therefore, I would like to invite you for minor revision that are mainly related to English language and grammar mistakes. please carefully address other issues related to your manuscript in the revised version of the manuscript

We look forward to receiving your revised manuscript.

Kind regards,

Aziz ur Rahman Muhammad

Academic Editor

PLOS ONE

Journal Requirements:

"This study was funded by the Coordenação de Aperfeiçoamento de Pessoal de Nível Superior (CAPES; Brazil; scholarship of master’s degree in Animal Science for Suziane Rodrigues Soares at the Universidade Federal de Mato Grosso – Campus Sinop; 2016-2018), Conselho Nacional de Desenvolvimento Científico e Tecnológico (CNPq, Brazil; Number: 309450/2019-5) and Ministério Público do Estado de Mato Grosso (Fundação Uniselva/UFMT. Brazil; Number SEI 23108.066569/2023-30). The funders were not involved in data or paper preparation."

Additional Editor Comments:

Dear Authors,

Reviewers has reviewed the manuscript and all of the 3 reviewers suggested minor revision. Therefore, I would like to invite you for minor revision that are mainly related to English language and grammar mistakes. please carefully address other issues related to your manuscript in the revised version of the manuscript

Reviewers' comments:

Reviewer's Responses to Questions

**Comments to the Author**

1. Is the manuscript technically sound, and do the data support the conclusions?

Reviewer #1: Yes

Reviewer #2: Yes

Reviewer #3: Partly

2. Has the statistical analysis been performed appropriately and rigorously? 

Reviewer #1: Yes

Reviewer #2: Yes

Reviewer #3: I Don't Know

3. Have the authors made all data underlying the findings in their manuscript fully available?

Reviewer #1: Yes

Reviewer #2: Yes

Reviewer #3: Yes

4. Is the manuscript presented in an intelligible fashion and written in standard English?

Reviewer #1: Yes

Reviewer #2: Yes

Reviewer #3: No

5. Review Comments to the Author

Reviewer #1: L34-35: perhaps reword to “We estimated the net requirements of P and Ca for maintenance from the intercept of a nonlinear equation between mineral intake and the sum of fecal and urinary excretions, which is an estimate of endogenous P and Ca losses.”

L40: “… ) using a 5-fold cross validation approach.”

L43: “… means, respectively.”

L46: “were” or “are”?

L46: “… for lactating dairy…”

L49: “… sustainability of the dairy…”

L62: define AC here at first use.

L64-66: “…, we propose here the term metabolizable coefficient as opposed to AC.”

L71: “… adopted to estimate endogenous…”

L91-92: abbreviate to MC

L98: abbreviate to MC

L107: delete one of the “P and Ca”

L109: this meta-analysis focused on P and Ca, but you only used search terms of “dairy cows” and “phosphorus”? Why wasn’t “calcium” used as a search term?

L110: was the second systematic review done using the same search terms?

L112: “review” instead of “revision” in all instances that it is used similarly.

L126: “reported in studies”

L126: “were” instead of “are”

L150: missing a period

L186: hasn’t MC already been defined?

L188-189: I recommend entering this equation using the “Equation” functionality of Microsoft word (under “Insert” and then “Equation”)

L198: should this be “studentized residual”?

L206: should this be “… using the 5-fold cross-validation approach…”?

L224: “genotype” or “breed”?

L226-227: was urine P and Ca exclusively measured by total collection? Did not all studies measure urine output by total collection? Why would they measure urine output by total collection and then use markers to assess fecal output in the same study?

L244-245: Did you describe in the methods how mean and slope biases were assessed? Also, I don’t know if slope being not different from 1 necessarily indicates linear bias. St-Pierre (2003; DOI: 10.3168/jds.S0022-0302(03)73612-1) suggests regressing residuals (obs – predicted) against mean centered predicted values, then the intercept of this regression is mean bias, and the slope is slope bias.

L307-308: was this a significant difference between Jersey and Holsteins? The standard errors make it seem as if they would not be statistically different.

L351: “… introduced the term…”

L354: “… the concept of metabolizable mineral aligns…”

L356-358: If MC and AC are so similar, what is the benefit of using MC over AC?

L364-365: “… is feasibly measured in a commercial…”

L424 “Keanthao et al. [50] reported…”

L427: “… to adequately estimate P…”

L434: “highlighted”

L436: “NASEM”

L436-437: reword.

L439: delete “be”

Reviewer #2: This meta-analysis developed and evaluated models to estimate Ca and P requirements for lactating dairy cattle using NL mixed models. Overall, the methodology is adequate, and the manuscript is well-written. The limitations of this study were well-discussed and justified. The parameter estimates had low standard errors, and cross-evaluation indicated opportunities for practical applications. Additionally, the authors recognized that an additional model evaluation using independent studies is recommended. For this reviewer, the major issue is to clarify if growing animals or late gestation animals were used in this study. If yes, the authors need to justify why they did not explore those requirements. Therefore, please find below some comments for consideration

Title: Use "A new system of phosphorus and calcium requirements for lactating dairy cows"

L33: Use “articles or studies” instead of “papers” across the entire manuscript.

L33: Please describe the database used in the abstract (breed, averages for DMI and milk yield x±xx).

L37: Use “treatment means”.

L40: “.. using a 5-fold cross-validation approach…” instead of “...from the..”.

L46: “lactating” instead of “lactation”.

L93: Remove the word "system" and use "model."

L107: Rewrite this sentence, something like: “A systematic review of mineral balance trials published as peer-review publications (xx studies) was performed. Treatment means were used to develop models to estimate P and Ca requirement for lactating dairy cows”.

L124: Remove this sentence “...(53 from previous review and four from new 124 revision).."

L126: Clarify this sentence “Data not reported on studies were evaluated as missing data; then, they are

subsequently excluded from the final model.” How was the evaluation conducted? Do the authors mean summarized using descriptive statistics?

Table 1: Confirm if that is the final database used to develop the models, if not, please describe only the final database used for model fitting. Also, describe the frequency for parity (primiparous, multiparous, or both) and fecal collection methods (total or estimated using markers) using the final database.

L139: Describe how many treatment means were dropped as outliers.

L149: Describe the N or % of treatment means that did not report SEM in your database. Include in the discussion section the methods used as weight in meta-analysis and why did the authors use SEM instead of other methods (i.e. N of animals / trt).

L153: What is the mature BW for each breed used in this database? Based on the minimum BW and maximum DIM, the authors need to discuss if some cows had growth/pregnancy requirements for Ca and P which were not explored in this study. If yes, include it in the limitation of study.

L159: Describe how the initial values for parameters of the NL mixed models were identified. Describe it.

L162: Check if the intercept β1 is significant in this model, include it in the results section.

L159: Explain why a nonlinear exponential mixed model was chosen over other models (i.e., linear mixed models with linear or quadratic terms, did you test it or that was due to data visualization?).

L140 and L177: Delete redundant information.

L66/98/186/L194: Revise abbreviation use for metabolizable coefficient (MC) across the entire manuscript.

L221: Clarify if some animals were growing in this database (316 to 754 kg).

L221: Add the DMI range.

L236: Remove the space before the comma.

L244: Use "slope bias" instead of “slop”.

L331: Include a discussion about outliers in meta-analysis here. Also, is the low milk yield due the diet? For example, treatment means from the pasture system had lower milk. Discuss it.

L334: Include a discussion about internal and external model evaluation. Also, discuss the % of the data (treatment means) that was from early, middle and late lactation. Also, the authors need to discuss the methods used to estimate Ca and P excretion in feces and their limitations (markers)..

L395: Include a period in this phrase “… ecosystems [3, 4]. Our…”.

L416-424: Fix parenthesis.

L436: Fix a typo “NSAEM (2021)”

L448/449: Explain the main limitations of cross-validation in terms of model evaluation.

Reviewer #3: Peer review - A new system of phosphorus and calcium requirements for dairy cows

Overall, this study has merit as it aims to develop a new model for predicting the calcium and phosphorous requirements of dairy cows. Previous requirements (NASEM and NRC) were predicted from equations derived from a rather small sample size which this study aims to overcome by including a far larger sample size in their regression analysis and subsequent prediction equations. It is not clear to the reader whether the studies used to create the NASEM and NRC predictions are also included in the present study (after checking the reference list and excel file, it seems they are (?)) – this should be highlighted as it adds merit and credibility to the present study.

I have answered No to question number 4 as the current manuscript requires English language editing to avoid ambiguities and improve understanding. Additionally, there are a number of typos throughout the manuscript that should be corrected before publishing e.g. L46, “lactation” should read “lactating”, L47, “requirement” should read “requirements”, L49, should read “in the dairy industry”… etc.

The introduction is well thought out and lays a solid foundation for the study. The discussion requires reformulation and rewriting.

Please find my specific comments below:

Comments:

The title does not adequately describe the manuscript. Consider revising. e.g. “A new system to predict phosphorous and calcium requirements in dairy cows”

The authors should ensure that any studies published after 6 November 2017 but before 1 January 2018 have not been accidently overlooked due to the nature of the follow up search term. I suggest that the year 2017 be included in the search term instead of the year 2018 and any duplicated studies in 2017 be removed.

Figure 1: Instead of using the headings “Identification of previous studies via databases” and “Identification of new studies via databases”, I would suggest the authors refer to these searches as initial and follow up searches. Dates could be included to avoid any ambiguity. Double check the use of brackets and spacings in Figure 1.

Please double-check the numbers in the screening section of the PRISMA diagram, there seem to be some inconsistencies?

The number of mineral balance trials included is not clear as the text and Table 1 state 76 trials however, there are only 72 trials included in the Excel file containing the complete dataset (reference 21), please clarify.

Figure 2: Please update this figure to include the word excretion in the y axis label e.g. Total P excretion… and Total Ca excretion… to avoid any confusion.

Figure 3: Please update this figure to include the word excretion in the x and y axes labels e.g. Observed total P excretion… and Observed total Ca excretion… to avoid any confusion. In Figure 3 it is stated that 5 folds were created by study grouping to guarantee independence between observed and predicted values. According to S2, the folds were created on Monday 6 September 2023 but the follow up literature search was conducted in March 2024 – I imagine that no new studies were published and included in the meta-analysis between these dates, however, this should be explicitly stated (in the text rather than in the figure legend) for clarity.

Table 2: Please include the word “analysis” in the heading for clarity i.e. Nonlinear mixed regression analysis…

S3: Please double check that these graphs are correct? If so, please label with a term other than “dotted lines” to improve clarity, I do not see any dotted lines(?).

Figure 4: What does the X mean in Figure 4? Please add to the figure legend for clarity.

Figure 5: Include the meaning of X for clarity i.e. X represents the mean concentration. Label Holstein and Jersey on the x axis to avoid any confusion.

Discussion

Figure 6: These are nice graphs that help the reader to visualize requirements predicted by different entities. Since the equations and absorption coefficients have been included for the other models, I suggest including the same for the authors proposed model. It is not entirely clear when looking at the figure that the authors are referring to 10kg milk yield/ 20/ 30 etc. Please include “milk” or “milk yield” or an abbreviation e.g. “MY” in the figure.

Lines 393 – 395: This is a repetition from the introduction as it is, however, it is a good point and should be expanded upon in the discussion.

Lines 415 – 416: This may be correct, however, it is not deducible from Figure 6 as there is no indication of DMI or the equation used to derive these values related to Figure 6? Please clarify either in the figure, the legend or in the text.

Line 421: When referring to dietary phosphorous or calcium intake, add the word “dietary” before P (or Ca) e.g. “dietary P” to avoid any ambiguity. Please check this throughout the manuscript.

Line 426: Are there really so few dose-response relationship studies on Ca and P in dairy cows? If so, I would make this very apparent in the introduction and then build on this in the discussion (as the authors have done to a certain degree), however, more compelling wording and insight is required for an improved discussion.

Lines 428 – 430: Please recalculate these percentages, there seems to be an error. Additionally, it would make it easier for the reader if the authors referred to a specific milk yield and then another e.g. “Our proposed model predicts a net calcium requirement 117% higher than the NASEM and 79% higher than the NRC recommendations for a 500kg dairy cow producing 10kg of milk per day. For cows producing 50kg (700kg) of milk a day, our model predicts….”

Lines 432 – 433: To enhance clarity rewrite in the same fashion as suggested above.

Lines 434 – 437 need to be rewritten to provide clarity.

Line 449 – 450: Mention which other factors are involved in endogenous fecal losses for completeness.

Line 453: Use “actual” for known MC instead of specific true

References:

Please use the same referencing style throughout. See references 10, 12, 16, 17 (I have not been through all the references) for examples where the referencing style deviates from the majority.

6. PLOS authors have the option to publish the peer review history of their article (what does this mean?). If published, this will include your full peer review and any attached files.

Reviewer #1: **Yes: **Matthew R. beck

Reviewer #2: No

Reviewer #3: No

---

## [Author Response · Author response to Decision Letter 0]

6 Jun 2024

Please find attached a file with Response to Reviewers

---

## [Decision Letter · Decision Letter 1]

1 Jul 2024

PONE-D-24-09154R1A new system of phosphorus and calcium requirements for lactating dairy cowsPLOS ONE

Dear Dr. Oliveira,

Thank you for submitting your manuscript to PLOS ONE. After careful consideration, we feel that it has merit but does not fully meet PLOS ONE’s publication criteria as it currently stands. Therefore, we invite you to submit a revised version of the manuscript that addresses the points raised during the review process.

Dear Authors with the reviewer comments, please also improve conclusion and provide recommendation for further research/meta-analysis in mineral in nutrition  Please submit your revised manuscript by Aug 15 2024 11:59PM. If you will need more time than this to complete your revisions, please reply to this message or contact the journal office at plosone@plos.org. Please include the following items when submitting your revised manuscript:A rebuttal letter that responds to each point raised by the academic editor and reviewer(s). You should upload this letter as a separate file labeled 'Response to Reviewers'.A marked-up copy of your manuscript that highlights changes made to the original version. You should upload this as a separate file labeled 'Revised Manuscript with Track Changes'.An unmarked version of your revised paper without tracked changes. You should upload this as a separate file labeled 'Manuscript'.If applicable, we recommend that you deposit your laboratory protocols in protocols.io to enhance the reproducibility of your results. Protocols.io assigns your protocol its own identifier (DOI) so that it can be cited independently in the future. For instructions see: https://journals.plos.org/plosone/s/submission-guidelines#loc-laboratory-protocols. Additionally, PLOS ONE offers an option for publishing peer-reviewed Lab Protocol articles, which describe protocols hosted on protocols.io. Read more information on sharing protocols at https://plos.org/protocols?utm_medium=editorial-email&utm_source=authorletters&utm_campaign=protocols.

We look forward to receiving your revised manuscript.

Kind regards,

Aziz ur Rahman Muhammad

Academic Editor

PLOS ONE

Journal Requirements:

Additional Editor Comments:

Dear Author

Please address the reviewer comments and improve you conclusion and provide recommendations for further research/meta-analysis in mineral nutrition for lactating cows.

Reviewers' comments:

Reviewer's Responses to Questions

**Comments to the Author**

1. If the authors have adequately addressed your comments raised in a previous round of review and you feel that this manuscript is now acceptable for publication, you may indicate that here to bypass the “Comments to the Author” section, enter your conflict of interest statement in the “Confidential to Editor” section, and submit your "Accept" recommendation.

Reviewer #1: All comments have been addressed

Reviewer #2: All comments have been addressed

Reviewer #3: All comments have been addressed

2. Is the manuscript technically sound, and do the data support the conclusions?

Reviewer #1: Yes

Reviewer #2: Yes

Reviewer #3: Yes

3. Has the statistical analysis been performed appropriately and rigorously? 

Reviewer #1: Yes

Reviewer #2: Yes

Reviewer #3: Yes

4. Have the authors made all data underlying the findings in their manuscript fully available?

Reviewer #1: Yes

Reviewer #2: Yes

Reviewer #3: Yes

5. Is the manuscript presented in an intelligible fashion and written in standard English?

Reviewer #1: Yes

Reviewer #2: Yes

Reviewer #3: No

6. Review Comments to the Author

Reviewer #1: I believe that this revised manuscript has adequately addressed my previous minor suggestions. I feel that all analysis and interpretations of their results are adequate.

Reviewer #2: Different statistical approaches, animal genotypes, climate, and research protocols change over time. All meta-analyses have limitations, but we need to clearly state them (statistical methods, software used, and data availability). One interesting discussion during the 2024 ADSA meeting this year was about variation in NDF analysis due to differences in lab analysis. I believe that this type of discussion is important for mineral nutrition too.

Over time, several research groups have been discussing how to improve our Dairy Science or Modeling Science. This reviewer made some points about methods for weighting studies in meta-analysis (SEM), nonlinear mixed-effects models, cross-validation, and other aspects to encourage the authors to discuss ways to improve our meta-analysis approach. Discussing the limitations and benefits of current meta-analysis methods and data available can prompt readers to think about these challenges, how this can affect the results, and address them in future research. A popular method widely used in meta-analysis doesn’t mean it cannot be improved.

This reviewer respectfully disagrees with SEM being considered a gold-standard method, although it is widely used in Animal Science meta-analysis. SE from GLM and Mixed models are fitted differently; SE from GLM is largely underestimated. If not evaluated carefully, this can cause under- or over-weighting in the model (see the discussion by Hanigan et al., 2021, 2024: https://doi.org/10.3168/jds.2020-19672 , https://doi.org/10.3168/jds.2024-24230 ). This reviewer agrees that this type of discussion is more appropriate for a statistical methods paper. However, if the authors choose to use this method, they need to justify it.

This reviewer appreciates the biology discussion, framework description, and codes, which are acceptable in science. The major questions for the conclusion section are: do we need a new system or do we need recent/new studies in mineral nutrition for lactating dairy cows? The authors can discuss the N of papers published in the last 5 years in their database. Also, please include recommendations for further research/meta-analysis.

Reviewer #3: Dear Authors, thank you for addressing my previous concerns to a large extent, however, there a still quite a few small grammatical errors that need to be addressed before publication. I’ve made a list of the most obvious ones below but please have a native English speaker go through your manuscript once again. I’ve advised to accept your manuscript once these minor issues have been addressed. Thank you for an interesting read.

L45: Add the word “the” before “metabolizable coefficient”

L71: Should read “mineral balance” instead of “balance mineral” trials…

L88: “model of” should read “models for”

L90: Use “requires development” instead of “is still necessary to be developed”

L94: “of” P and Ca… should read “for”

L95: “nutrients” should be singular

L109: “requirement” should be plural

L118 and L121: “peer review” should read “peer reviewed”

L218: Add the word “an” before “iterative approach”

L236: “of” should be “from”

L270: Remove the word “cows” from this sentence

L272: “of” should read “from”

L281: “cow” should be singular and “dataset” plural

L282-283: Do you mean the parity was not reported in 34 studies? If so, please change the wording of this sentence.

Table 3: Do you mean “Dietary starch” by “Starch diet?” If you use “Dietary starch” or simply “Starch”

L390: The fullstop (period) should be a comma in this sentence

L407: “mechanism” should be plural

L419: Here “metabolizable” should read “metabolizability”

L427: “the” should read “that”

L499: “to adequately” instead of “adequately to”

L501: “requirements” should be singular

L502: remove “of Ca”

L503: First “cows” should be singular

L509: “requirements” should be singular

L512: “requirements” should be singular

L513: Replace “Because od” with “Due to the”

L527: “sampling spot” should read “spot sampling”

L527: “are evidences” should read “is evidence”

L529: “to” should read “for”

L530: “collect” should read “collection”

Figure 1: Missing opening brackets for “Phosphorus”. Missing closing bracket in first box. Please revise.

Figure 4 has not been updated. My previous comment: Figure 4: What does the X mean in Figure 4? Please add to the figure legend for clarity.

7. PLOS authors have the option to publish the peer review history of their article (what does this mean?). If published, this will include your full peer review and any attached files.

Reviewer #1: No

Reviewer #2: No

Reviewer #3: No

---

## [Decision Letter · Decision Letter 2]

1 Aug 2024

A new system of phosphorus and calcium requirements for lactating dairy cows

PONE-D-24-09154R2

Dear Dr. Oliveira,

We’re pleased to inform you that your manuscript has been judged scientifically suitable for publication and will be formally accepted for publication once it meets all outstanding technical requirements.

Kind regards,

Aziz ur Rahman Muhammad

Academic Editor

PLOS ONE

Additional Editor Comments (optional):

Dear Authors

Thanks for considering the comments of reviewers. Good Luck

Reviewers' comments:

Reviewer's Responses to Questions

**Comments to the Author**

1. If the authors have adequately addressed your comments raised in a previous round of review and you feel that this manuscript is now acceptable for publication, you may indicate that here to bypass the “Comments to the Author” section, enter your conflict of interest statement in the “Confidential to Editor” section, and submit your "Accept" recommendation.

Reviewer #2: All comments have been addressed

2. Is the manuscript technically sound, and do the data support the conclusions?

Reviewer #2: Yes

3. Has the statistical analysis been performed appropriately and rigorously? 

Reviewer #2: Yes

4. Have the authors made all data underlying the findings in their manuscript fully available?

Reviewer #2: Yes

5. Is the manuscript presented in an intelligible fashion and written in standard English?

Reviewer #2: Yes

6. Review Comments to the Author

Reviewer #2: (No Response)

7. PLOS authors have the option to publish the peer review history of their article (what does this mean?). If published, this will include your full peer review and any attached files.

Reviewer #2: No

---

## [Editor Report · Acceptance letter]

20 Aug 2024

PONE-D-24-09154R2 

PLOS ONE

Dear Dr. Oliveira, 

I'm pleased to inform you that your manuscript has been deemed suitable for publication in PLOS ONE. Congratulations! Your manuscript is now being handed over to our production team.

Kind regards, 

on behalf of

Dr. Aziz ur Rahman Muhammad 

Academic Editor

PLOS ONE